

# Modifying effects of leaf litter extracts from invasive versus native tree species on copper-induced responses in *Lemna minor*

Rolandas Karitonas[1,*], Sigita Jurkonienė[1], Kazys Sadauskas[1], Jūratė Vaičiūnienė[2] and Levonas Manusadžianas[1,*]

[1] Institute of Botany, Nature Research Centre, Vilnius, Lithuania
[2] Institute of Chemistry, Centre for Physical Sciences and Technology, Vilnius, Lithuania
[*] These authors contributed equally to this work.

## ABSTRACT

Invasive plant species tend to migrate from their native habitats under favourable climatic conditions; therefore, trophic and other relationships in ecosystems are changing. To investigate the effect of natural organic matter derived from native *Alnus glutinosa* tree species and from invasive in Lithuania *Acer negundo* tree species on copper toxicity in *Lemna minor*, we analysed the dynamics of Cu binding in aqueous leaf litter extracts (LLE) and plant accumulation, morphophysiological parameters, and antioxidative response. The results revealed that *A. glutinosa* LLE contained polyphenols (49 mg pyrogallol acid equivalent (PAE)/g DM) and tannins (7.5 mg PAE/g DM), while *A. negundo* LLE contained only polyphenols (23 mg PAE/g DM). The ability of LLE to bind Cu increased rapidly over 1.5–3 h to 61% and 49% of the total Cu concentration ($6.0 \pm 0.9$ mg/L), respectively for *A. glutinosa* (AG) and *A. negundo* (AN), then remained relatively stable until 48 h. At the same time, *L. minor* accumulated 384, 241 or 188 $\mu$g Cu/g FW when plants were exposed to Cu (100 $\mu$M $CuSO_4$), Cu with 100 mg/L dissolved organic carbon (DOC) from either AG LLE or AN LLE, accordingly. Catalase (CAT) and guaiacol peroxidase (POD) played a dominant role in hydrogen peroxide scavenging when plants were exposed to Cu and 10 or 100 mg/L $DOC_{AG}$ mixtures in both the first (up to 6h) and the second (6–48 h) response phases. Due to functioning of oxidative stress enzymes, the levels of the lipid peroxidation product malondialdehyde (MDA) reduced in concentration-dependent manner, compared to Cu treatment. When combining Cu and $DOC_{AN}$ treatments, the most sensitive enzymes were POD, ascorbate peroxidase and glutathione reductase. Their activities collectively with CAT were sufficient to reduce MDA levels to Cu-induced in the initial, but not the second response phase. These data suggest that leaf litter extracts of different phenolic compositions elicited different antioxidant response profiles resulting in different reductions of Cu stress, thus effecting *L. minor* frond and root development observed after seven days. The complex data from this study may be useful in modelling the response of the aquatic ecosystem to a changing environment.

Corresponding author
Rolandas Karitonas,
rolandas.karitonas@gamtc.lt

## INTRODUCTION

Copper is needed for normal plant growth and development and is a cofactor for physiological processes such as photosynthesis, mitochondrial respiration, superoxide scavenging, ethylene sensing and lignification (*Maksymiec, 1997*). However, copper released into the environment in surplus concentrations is toxic to plants (*Naumann, Eberius & Appenroth, 2007*). Transition metals, including Cu, stimulate the formation of hydroxyl radicals ($\cdot$OH) from the non-enzymatic chemical reaction between superoxide ($O_2^-$) and $H_2O_2$ (Haber-Weiss reaction). Excess Cu can induce negative effects including the production of reactive oxygen species (ROS) via Fenton reaction (*Halliwell & Gutteridge, 1984*). ROS, in turn, can oxidize lipids (*De Vos et al., 1991*), disrupt protein functions due to binding to sulphydryl groups (*Weckx & Clijsters, 1996*) and inhibit photosynthesis and electron transport (*Thomas et al., 2013*; *Xia & Tian, 2009*). Plant growth can be inhibited as a result.

The presence of oxygen in intracellular environments due to aerobic metabolism poses a constant oxidative threat to cellular structures and processes. ROS affect the metabolism, growth and development of plant cells. ROS formation and consumption are tiny balanced and coherent in cells. At any condition in which cellular redox homeostasis is disrupted, ROS production becomes far greater than the capacity of the tissues to scavenge them, thus can be defined as oxidative stress (*Alscher, Donahue & Cramer, 1997*). The environmental stressors can increase the synthesis of non-enzymatic antioxidants such as thiol tripeptides, glutathione and ascorbate and $\alpha$-tocopherol, as well as the modification of the activity of antioxidant enzymes including superoxide dismutase, catalase, glutathione peroxidase, ascorbate peroxidase and glutathione reductase (*Foyer et al., 1997*; *Schützendübel & Polle, 2002*). Accordingly, excess of Cu in plants can cause oxidative stress, and therefore change antioxidative pathways (*Babu et al., 2003*; *De Vos et al., 1992*; *Gupta et al., 1999*; *Teisseire & Guy, 2000*; *Wang et al., 2004*).

The solubility, adsorption, transport and toxicity of metals in natural surface waters are strongly influenced by complexation with dissolved organic matter (DOM) (*Kim et al., 1999*; *Koukal et al., 2003*; *Manceau & Matynia, 2010*; *Marx & Heumann, 1999*). Natural waters contain different concentrations of DOM, which, depending on the geographical area are ranging between < 1 and 100 mgC/L (*Wetzel, 2001*). Higher DOM concentrations (up to 300 mgC/L) have also been reported in some Canadian wetlands (*Blodau, Basiliko & Moore, 2004*). DOM is a complex and polymorphous mixture, which includes proteins, carbohydrates, polyphenols, and other vital compounds that originate chiefly from the degradation of plant and animal matter (*Stevenson, 1994*). A major fraction of DOM in waters comprises humic substances representing more than 60 −80% of the total dissolved organic carbon (DOC), which consists mainly of humic and fulvic acids (*Steinberg, 2003*).

Leaf litter is a readily available allochthonous source of DOM and plays an important role in freshwater ecosystems serving as a key source of nutrients (*Tank et al., 2010*). The enrichment with DOM relates to litter quality, which, in turn, is leaf species-dependent and furthermore can depend whether they are of native or non-native origin (*Casas et al., 2013*). In addition, leaf litter emits phenolic compounds including tannins, a significant

component of plant secondary metabolites (*Lachman et al., 2011*; *Lin et al., 2006*). Tannins may provide a nutrient conservation mechanism by reducing decomposition rates of litter and decreasing nitrogen leaching potential (*Lin et al., 2010*). It has been found that DOM might cause oxidative stress in freshwater organisms (*Nimptsch & Pflugmacher, 2008*; *Steinberg et al., 2003*), and leaf litter leachates obtained from various tree species such as white pine *Pinus strobus* and red oak *Quercus robur* (*Earl, Cohagen & Semlitsch, 2012*) can be toxic to aquatic organisms. Moreover, it has been suggested that black alder *Alnus glutinosa*, native in Lithuania species, and boxelder maple *Acer negundo,* invasive in Lithuania species, impacted the same aquatic organisms in different ways (*Krevš et al., 2013*; *Manusadžianas et al., 2014*). *A. negundo* became widespread in Lithuania after its escape from cultivation in the mid-twentieth century (*Gudžinskas, 1998*). It colonized coastal zones of lakes and rivers that are dominated by autochthonous *A. glutinosa* (*Prieditis, 1997*). In this context, it might be interesting to reveal the potential of the DOM obtained from diverse species to modify metal effects on aquatic plants.

Alongside the natural sources, the increase of DOM in water bodies depends on rural and municipal activities. Similar anthropogenic sources of copper appearance were emphasized, i.e., agriculture and industrial wastes (*Hou et al., 2007*; *Panagos et al., 2018*), and combating massive growth of cyanobacteria (*Huh & Ahn, 2017*). To investigate possible phytotoxicity effects, we used duckweed *Lemna minor*, a well-known bioindicator of eutrophic water bodies (*Environment Canada, 2007*; *US Environmental Protection Agency, 2012*). This plant is considered to be a suitable model for physiological and ecotoxicological studies due to its small size, fast growth rate, vegetative reproduction, ease of culture and sensitivity to numerous pollutants. *Lemna* has been used for antioxidative response studies (*Forni et al., 2012*; *Radić & Pevalek-Kozlina, 2010*; *Razinger et al., 2007*; *Teisseire, Couderchet & Vernet, 1998*; *Teisseire & Guy, 2000*; *Zezulka et al., 2013*). However, information on the involvement of oxidative stress under the combined treatment of Cu and DOM obtained from leaf litter extracts is lacking in *L. minor*.

The main objective of this study was to investigate the effect of natural organic matter derived from native *Alnus glutinosa* tree species and from invasive in Lithuania *Acer negundo* tree species on copper toxicity in *L. minor*. We focused on time-dependent alterations of (1) Cu binding to DOM in media and accumulation in the plant, (2) morphophysiological parameters (frond area and root length), and (3) oxidative stress characteristics such as lipid peroxidation, hydrogen peroxide content and the activities of antioxidant enzymes, i.e., catalase, guaiacol and ascorbate peroxidases, and glutathione reductase. We limited our observations of oxidative stress characteristics to 48 h; thus, relatively high concentrations of 100 $\mu$M CuSO$_4$ and up to 100 mg/L DOC were used.

## MATERIAL AND METHODS

### Plant material

We have been collecting duckweed (*Lemna minor* L.) plants from a local freshwater pond (54°75′N, 25°29′E; Verkiai Regional Park, Vilnius) and cultured fronds under controlled conditions for several years. Species specificity of experimental clone morphologically

similar to *L. minor* was proved by sequencing the chloroplast DNA fragment (611 bp) at the Laboratory of Molecular Ecology of the Nature Research Centre (Vilnius). The highest chloroplast DNA sequence identity (99%) to *L. minor* strain RDSC 7210 (Sequence ID KX212888.1), which included ATPase subunit I gene, partial cds; atpF-atpH intergenic spacer, complete sequence; and ATPase subunit II gene, partial cds, indicated that plant clones used for the experiments in the current study should be attributed to *L. minor* species.

A stock culture was cultivated in 500-mL plastic containers $60 \times 85 \times 100$ (mm, WxLxH) in a modified Steinberg growth medium containing 3.46 mM $KNO_3$, 1.25 mM $Ca(NO_3)_2$, 0.66 mM $KH_2PO_4$, 0.072 mM $K_2HPO_4$, 0.41 mM $MgSO_4$, 0.63 μM $ZnSO_4$, 1.94 μM $H_3BO_3$*, 0.18 μM $Na_2MoO$*, 0.91 μM $MnCl_2$*, 2.81 μM $FeCl_3$* and 4.03 μM $Na_2EDTA$** (chemicals were from Roth, Acros Organics* and Fisher**). The pH was adjusted to 6.0 with 1 M NaOH.

Stock cultures and treated plants were kept in growth chambers at 24 $\pm$2 °C under constant illumination with cool white fluorescent light at a photosynthetic photon flux density of 160μmol m$^{-2}$ s$^{-1}$.

## Preparation of aqueous extracts

Fallen leaves of *A. negundo* and *A. glutinosa* were collected in Rudnia (54°05′N, 24°40′E; Varena district, Lithuania) in autumn. After collection, tree leaves were dried for 10 days in the shade at room temperature. The dried materials (without petiole and central vein) were mechanically ground to obtain a homogenous powder. One gram powder was extracted in 100 mL of deionised water (dH$_2$O) for 3 h at 65 °C followed by rapid filtration through a Whatman #3 disk in order to obtain a clear crude extract solution, and then re-filtered through a nitro-cellulose paper filter (0.2 μm) to reduce the risk of interference by microorganisms.

## Experimental design

The experimental scheme comprised control (growth medium) and four treatments (growth medium supplemented with Cu (100μM $CuSO_4$ or 6.4 mg Cu/L); 100 mg/L DOC; Cu + 10 mg/L DOC and Cu + 100 mg/L DOC. Plants were incubated in 500-mL plastic containers ($60 \times 85 \times 100$) having 200 mL of the corresponding medium. Media were adjusted to pH 6.

For growth experiments, healthy colonies with 2–3 fronds from stock cultures were transferred to the containers with the corresponding exposure medium. The frond area (zones without the signs of chlorosis) and root length were measured at 0-day and after seven days by using image control system (Software MOTIC 2.0). The growth rate per day was calculated with the following equation $r = (\ln x_{t2} - \ln x_{t1})/t_2 - t_1$, where $x_{t1}$ and $x_{t2}$ are the values of observation parameter at $t_1$ and $t_2$ day, respectively. Two independent experiments in quadruplicate were conducted.

To study $H_2O_2$ kinetics, lipid peroxidation and antioxidant enzyme activities, the cultures were started by transferring healthy colonies with 3–4 fronds from stock cultures into four containers (0.8–0.9 g in each container) for each corresponding exposure medium. Plant samples were collected 0.75, 1.5, 3, 6, 12, 24 and 48 h after the onset of the exposure. For

each exposure time, we prepared new exposure medium. Two independent experiments in quadruplicate for each parameter at each exposure time were conducted. Then samples of *L. minor* (0.2 g each) were processed after *Hildebrandt et al. (1986)*. Cold potassium phosphate buffer (0.1 M, pH 7.0) containing 1% (w:v) polyvinylpyrrolidone and 1% (v:v) Triton X-100 was added to chilled (4 °C) mortar and pestle containing the sample. Each sample was macerated with one mL of extracting buffer and was further grounded with another one mL of the buffer. A 1.5 mL aliquot of homogenate was centrifuged at 15,000 g for 15 min at 4 °C (*Hou et al., 2007*). The supernatant was frozen immediately for future total protein content and enzyme assays. Total protein content was determined using bovine serum albumin (BSA) as standard (*Bradford, 1976*). Spectrophotometric measurements were carried out on a Libra S32 PC UV–VIS (Biochrom, UK).

Chemicals used for the determination of the total protein content and oxidative enzyme activities were purchased from Sigma-Aldrich (ascorbic acid, BSA, $Na_2EDTA$, Folin-Ciocalteau's phenol reagent, hide powder, $H_2O_2$, pyrogallol, polyvinylpyrrolidone, sodium carbonate decahydrate, Triton X-100, thiobarbituric acid (TBA), trichloroacetic acid (TCA)) and from Roth (GSSG, $KH_2PO_4$, $K_2HPO_4$, NADPH, TRIS).

## Determination of Cu

Before determining the metal content, the plants from control and Cu, Cu + 10 mg/L DOC and Cu + 100 mg/L DOC treatments (one experiment with four replicates at each exposure time) were washed triple with deionised water. All liquids on the surface of plant materials were blotted with paper towels. 0.2 g of fresh weight of plant materials was transferred to a ceramic crucible (Haldenwanger, Waldkraiburg, Germany) to destroy the combustible (organic) portion of the sample by thermal decomposition in a muffle furnace (SNOL-1,6.2,5, Borispol, Ukraine) at 450–550 °C for 2–3 h. After the sample was digested in 0.5 mL pure $HNO_3$ (Roth, Karlsruhe, Germany) and heated until the acid evaporates up to a half volume and made up to final volume of five mL with $dH_2O$.

Cu fractionation in treatment solutions (without plants) followed *Adam et al. (2014)* procedure. The fraction with dissolved Cu was obtained after ultrafiltration for 1 h in Microsep[TM]Advance Centrifugal Devices (Pall Corporation, Ann Arbor, MI, USA) containing polyethersulfone membranes with a cut-off of 1 kDa, at 5,000 g (5430R, Eppendorf, Hamburg, Germany). Cu content was determined in duplicate at 0.75, 1.5, 3, 6, 12, 24 and 48 h before and after ultrafiltration by Perkin Elmer Optima 7000 Dual View ICP Optical Emission Spectrometer (Waltham, MA, USA) with standard method and calculated according to the standard curve. Cu concentration was expressed as mg/L or µg/g fresh weight (FW).

$CuSO_4$ $5H_2O$ was from Sigma-Aldrich (purum p.a.).

## Catalase (CAT) EC1.11.1.6

CAT was determined according to *Aebi (1984)*. The assay medium contained 50 mM potassium phosphate buffer (pH 7.4, 25 °C), 12.5 mM $H_2O_2$, 50 µl supernatant containing enzyme extract and $dH_2O$ to make up the volume to three mL. The reaction was initiated by adding $H_2O_2$. The decrease in absorbance of $H_2O_2$ was recorded at 240 nm for 60 s with

50 mM potassium phosphate buffer used as the blank. The enzyme activity was calculated from the initial rate of the reaction using extinction coefficient $\varepsilon = 0.04$ mM$^{-1}$ cm$^{-1}$ for $H_2O_2$.

### Ascorbate peroxidase (APX) EC 1.11.1.11

APX activity was measured according to the method described by *Nakano & Asada (1981)*. The three mL reaction medium was composed of 50 mM potassium phosphate buffer (pH 7.0, 25 °C), 0.5 mM ascorbic acid, 0.1 mM Na$_2$EDTA, 0.1 mM $H_2O_2$ and 100 μL supernatant containing enzyme extract. The reaction was initiated by adding $H_2O_2$. The decrease in the optical density to ascorbic acid was recorded at 290 nm for 30 s. The enzyme activity was calculated from the initial rate of the reaction using $\varepsilon = 2.8$ mM $^{-1}$ cm$^{-1}$ for ascorbate.

### Guaiacol peroxidase (POD) EC 1.11.1.7

POD activity was determined according to *Upadhyaya et al. (1985)*. The assay medium contained 2.5 mL of 50 mM potassium phosphate buffer (pH 6.1, 25 °C), one mL 1% $H_2O_2$, one mL 1% guaiacol and 50 μL supernatant containing enzyme extract. The reaction was initiated by adding supernatant containing enzyme extract. The change in the optical density was recorded at 420 nm for 1 min. The enzyme activity was calculated using $\varepsilon = 26.6$ mM$^{-1}$ cm$^{-1}$ for oxidized tetraguaiacol polymer.

### Glutathione reductase (GR) EC 1.6.4.2

GR activity was determined according to *Mannervik (1999)*. The assay medium contained 0.2 M potassium phosphate buffer (pH 7.0)/2 mM Na$_2$EDTA, 20 mM GSSG, 2 mM NADPH and 100 μL supernatant containing enzyme extract and dH$_2$O to make up the volume to one mL. The reaction mixture was equilibrated at 30 °C. The decrease in the NADPH concentration was recorded at 340 nm for 1 min against the assay solution. Corrections were made for the non-enzymatic oxidation of NADPH by recording the decrease at 340 nm without adding GSSG to the assay medium. The enzyme activity was calculated from the initial rate of the reaction after subtracting the non-enzymatic oxidation using $\varepsilon = 6.2$ mM$^{-1}$ cm$^{-1}$ for NADPH.

### Determination of hydrogen peroxide

The level of $H_2O_2$ in plant was determined according to *Jana & Choudhuri (1981)* with slight modification (*Chen, Lin & Kao, 2000*). $H_2O_2$ was extracted by homogenizing 0.2 g plant with 2 ml of phosphate buffer (50 mM, pH 7.0). The homogenate was centrifuged at 6,000 g for 25 min at 4 °C. 900 μL of the supernatant was mixed with 300 μL of 0.1% titanium chloride in 20% (v/v) $H_2SO_4$ and mixture was then centrifuged at 6,000 g for 15 min. The intensity of yellow colour of the supernatant was measured at 410 nm. $H_2O_2$ level was calculated using the extinction coefficient 0.28 μmol$^{-1}$cm$^{-1}$.

### Determination of lipid peroxidation

The level of lipid peroxidation in plant was assessed by thiobarbituric acid (TBA) reactive metabolites chiefly malondialdehyde (MDA) as described by *Heath & Packer (1968)*. Plant tissues (0.2 g) were extracted in two mL of 0.25% TBA made in 10% TCA. Extract was
heated at 95 °C for 30 min and then quickly cooled on ice. After centrifugation at 10 000 g for 10 min, the absorbance of the supernatant was measured at 532 nm. Correction of non-specific turbidity was made by subtracting the absorbance value taken at 600 nm. The level of lipid peroxidation was expressed as MDA concentration formed using $\varepsilon = 155$ mM$^{-1}$ cm$^{-1}$.

## Total phenol and tannin contents

The total phenol content of the dry leaves was measured spectrophotometrically after reaction with Folin-Ciocalteau phenol reagent, according to the manual method described by *Singleton, Orthofer & Lamuela-Raventós (1998)* with little modifications (*Amorim et al., 2008*; *Atanassova, Georgieva & Ivancheva, 2011*). A one mL aliquot of extracts or a standard solution of pyrogallol acid was added to a 25 mL volumetric flask containing nine mL of dH$_2$O. One mL of the Folin-Ciocalteau phenol reagent was added to the mixture and shaken. After 5 min, 10 mL of 7% Na$_2$CO$_3$ solution was added to the mixture. The solution was diluted to 25 mL with dH$_2$O and incubated for 30 min at room temperature. Absorbance was measured at 760 nm against a blank prepared with dH$_2$O.

For determination of tannin content, aqueous extracts of leaves were shaken with hide powder for 60 min in ultrasonic bath. The non-tannin phenolics in the clear supernatant were determined in the way similar to this of total phenol content. Tannin content was calculated as a difference between total phenolic and non-tannin phenolic contents in the extract. Pyrogallol acid in deionised water was used for making a standard curve. Total phenol and tannin values are expressed as pyrogallol acid equivalents (PAE) in mg/g DM.

## Dissolved organic carbon

DOC concentration in DOM extracts was determined according to ISO 8245:1999 in a certified analytical laboratory (JSC Water Investigations, Vilnius, Lithuania).

## Statistical analysis

The statistical analysis was carried out using the software PASW Statistics 18.0 (Predictive Analytics Software, IBM).

To validate an aggregation of the replicates from two experiments on frond and root growth rates, two-way MANOVA was used. The factors in the analysis were a two-level experiment factor (experiment 1 and experiment 2) and a seven-level treatment factor. To validate an aggregation of the replicates from two experiments on specific enzyme activity, MDA or H$_2$O$_2$ concentrations, two-way ANOVA was used. The factors in the analysis were a two-level experiment factor (experiment 1 and experiment 2) and a five-level treatment factor. In both analyses, there was no significant difference between two levels of experiment factor and interaction between the factors ($p > 0.05$), therefore, the replicates were pooled to yield $n = 8$.

After checking for normality (Shapiro–Wilk test) and homogeneity of variances (the Levene test) the differences of treatments from control within each leaf species were analysed by one-way ANOVA and the Dunnet test ($\alpha = 0.05$). Additionally, Tukey post-hoc test was used for differences among treatments ($\alpha = 0.05$).

## RESULTS

### Frond and root growth rate

The duckweed frond and root growth rates (FGR and RGR, respectively) were significantly affected by all treatments, with an exception of FGR in the treatment with 100 mg/L DOC derived from *A. negundo* (100-DOC$_{AN}$) (Fig. 1). Irrespective of the kind of leaf litter extract (LLE) and the type of treatment, the reduction in root length was higher than in the frond area.

As for the effect on fronds, Cu (100 µM or 6.4 mg/L) slowed the development of frond area by 97% after seven-day exposure (Fig. 1A). Due to the development of chlorosis and thus the decrease of a photosynthetically active square, the FGR was negative ($-0.3$ mm$^2$ d$^{-1}$). The 100-DOC$_{AN}$ diminished insignificantly the development of the frond area. Due to chlorosis, the FGR was $-0.02$ mm$^2$ d$^{-1}$ in Cu + 10-DOC$_{AG}$ treatment, and 5.0 times lower than the controls in Cu + 100-DOC$_{AG}$. The mixtures of Cu with *A. negundo* LLE of 10 and 100 mg/L DOC both stopped the growth of fronds.

As for the effect on roots, the growth was not observed over seven days in *L. minor* treated by Cu (Fig. 1B). 100 mg/L DOC derived from *A. glutinosa* (100-DOC$_{AG}$) significantly diminished the development of roots (root growth rate (RGR) decreased 2.1 times compared to controls). The 100-DOC$_{AN}$ significantly decreased the RGR by 1.6 times. The mixture of Cu + 10-DOC$_{AG}$ inhibited growth of the roots totally, while the effect of Cu + 100-DOC$_{AG}$ was lower (RGR decreased 5.1 times compared to controls). The mixtures of Cu with *A. negundo* LLE of 10 and 100 mg/L DOC both diminished RGR by 8.8 and 2.1 times compared to controls, respectively.

### Total phenols and tannins

Polyphenol and tannin concentrations in the DOM of *A. glutinosa* leaf litter were quite high (49 and 7.5 mg PAE/g DM, respectively), meanwhile DOM of *A. negundo* leaf litter had as much as twice lower polyphenol concentration (23 mg PAE/g DM) and no tannins.

### Cu accumulation

The ability of LLE to bind Cu increased rapidly. Lower concentrations of dissolved Cu were measured (after ultrafiltration) in the mixtures of Cu + 100-DOC$_{AG}$ and Cu + 100-DOC$_{AN}$ already after 45 min, making up 61.9% and 63.8% of the total Cu ($6.0 \pm 0.9$ mg/L, $n = 26$), respectively. The percent of dissolved Cu decreased further until 1.5 h in the case of 100-DOC$_{AG}$ and then remained relatively stable (39.3% of the total Cu) up to the end of exposure at 48 h, while it diminished until 3 h in the case of 100-DOC$_{AN}$ and then remained relatively stable up to the end of exposure (51.4% of the total Cu).

The Cu concentration in *L. minor* increased with time in all treatments (Fig. 2). The plant accumulated $99.6 \pm 4.6$ and $384 \pm 20.4$ µg Cu/g FW in the treatment with Cu, respectively at 45 min and 48 h. Similar Cu uptake was found in plants treated with Cu + 10-DOC$_{AG}$ or Cu + 10-DOC$_{AN}$. However, 100 mg/L DOC obtained from each of the leaf extracts diminished Cu uptake compared to Cu-treatment throughout all the exposure periods (Fig. 2). Accumulated Cu content was $55.6 \pm 2.7$ and $241 \pm 6.1$ µg Cu/g FW,

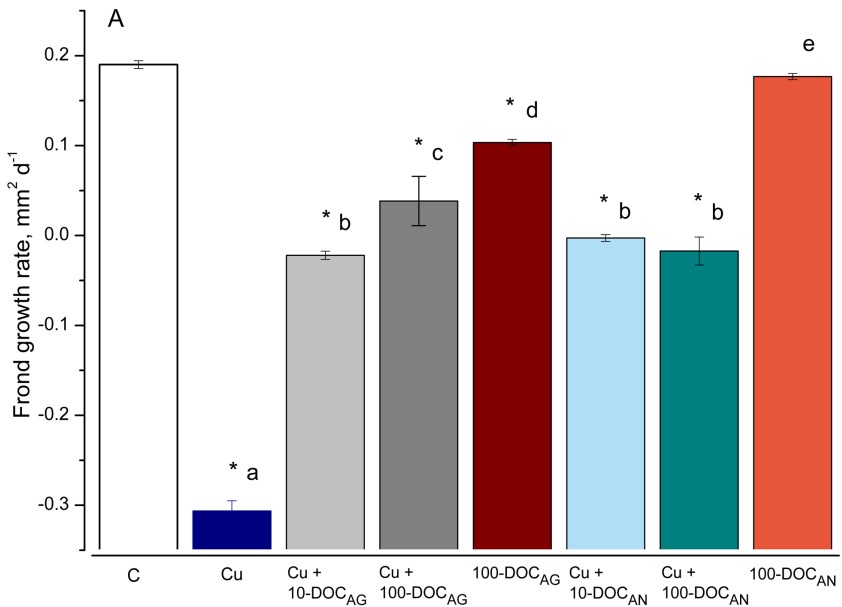

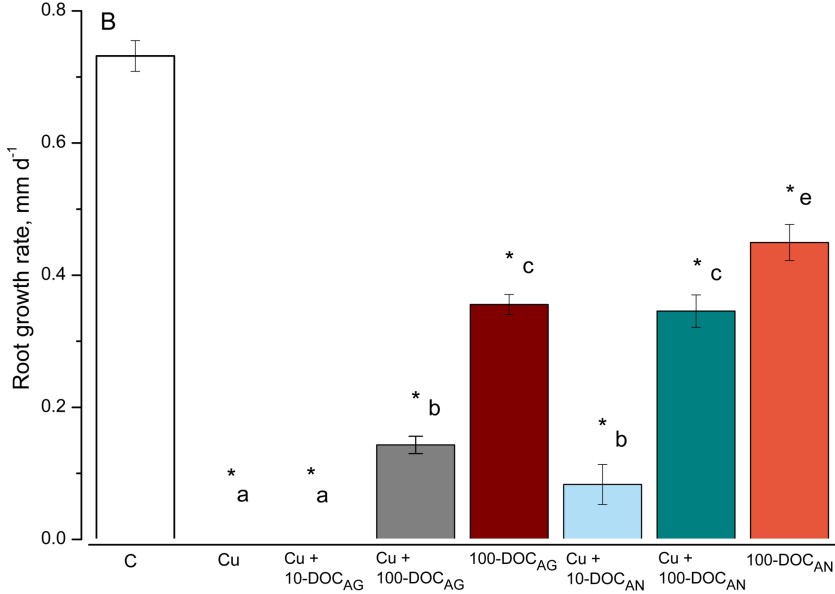

**Figure 1** **Frond and root growth of _L. minor_ exposed to copper, leaf litter extracts (LLE) of _Alnus glutinosa_ or _Acer negundo,_ and combinations of Cu and LLE.** Frond (A) and root (B) growth rates of _L. minor_ in control (C) and exposed for seven days with 100 μM $CuSO_4$ (Cu), mixtures of Cu and _A. glutinosa_ leaf litter extracts (LLE) of 10 and 100 mg/L DOC (Cu+10-DOC_AG and Cu+100-DOC_AG), mixtures of Cu and _A. negundo_ LLE of 10 and 100 mg/L DOC (Cu+10-DOC_AN and Cu+100-DOC_AN) as well as with 100 mg/L DOC of _A.glutinosa_ (100-DOC_AG) and _A. negundo_ (100-DOC_AN) LLE. Data represent mean ± SD ($n = 8$). Asterisks indicate significant difference from the control, different letters indicate significant difference among the means ($\alpha = 0.05$).

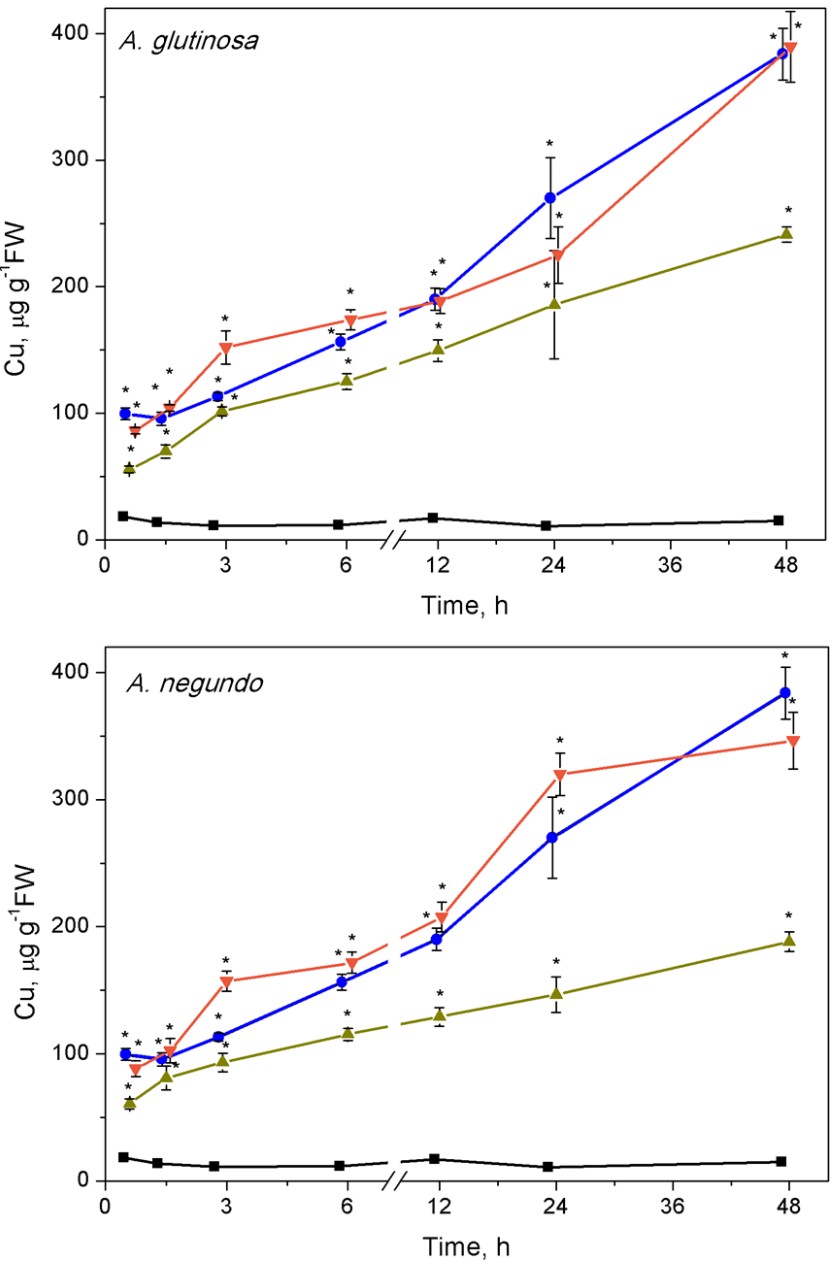

**Figure 2  Cu accumulation in *L. minor* treated by Cu, and combinations of Cu and leaf litter extracts from *A. glutinosa* or *A. negundo*.** Cu concentration in *L. minor* treated by Cu and Cu+DOC of leaf litter extracts ($\mu$g g$^{-1}$ FW). Plants were incubated in control medium (■), 100 $\mu$M Cu (●), 100 $\mu$M Cu+10 mg/L DOC (▼), 100 $\mu$M Cu + 100 mg/L DOC (▲) from *A. glutinosa* or *A. negundo*. Each value represents mean ±SD ($n = 4$). Asterisks indicate significant difference from the control at $\alpha = 0.05$.

respectively at 45 min and 48 h for *A. glutinosa*, and $60.8 \pm 4.0$ and $188 \pm 7.8$ $\mu$g Cu/g FW, respectively at 45 min and 48 h for *A. negundo* treatments.

## Hydrogen peroxide

The highest changes in hydrogen peroxide levels in plants were induced at early stages of the treatments. The $H_2O_2$ content increased significantly after 3 h of exposure to Cu, its combinations with either *A. glutinosa* or *A. negundo* extracts, and to 100-DOC$_{AG}$ or 100-DOC$_{AN}$ alone (Fig. 3). At this exposure time, it was approximately by 2.2 times higher in Cu, Cu+10-DOC$_{AG}$, Cu+100-DOC$_{AG}$ and 100-DOC$_{AG}$ treatments than in control (0.27 $\pm$ 0.10 mM, $n = 56$), and there were no significant differences among these treatments. However, treatments that included *A. negundo* extracts induced DOC–dependent $H_2O_2$ levels in *L. minor*. The $H_2O_2$ levels in Cu+10-DOC$_{AN}$ and Cu+100-DOC$_{AN}$ treatments exceeded that of Cu treatment by 21% and 30%, respectively, and reached maximum 40% value in the treatment of 100 mg/L DOC$_{AN}$.

At 12 h exposure, irrespective to the kind of leaf extract, the level of hydrogen peroxide in the treatments of Cu and its combinations with the extracts did not exceed that of control, but it was by 60% higher in the treatments of the leaf extracts alone (Fig. 3). Subsequently, within the 24–48 h exposure period, the level of $H_2O_2$ in *L. minor* treated with 100-DOC$_{AG}$ or 100-DOC$_{AN}$ decreased to that of control, whereas Cu and its combinations with leaf extracts induced augmentation of hydrogen peroxide up to 1.5–1.7 times.

## Lipid peroxidation

After 1.5 h of exposure, a significant increase of MDA concentration in *L. minor* over the control levels (Fig. 4, 10.1 $\pm$ 2.13 nM/g FW and 9.58 $\pm$ 2.22 nM/g FW, left and right graphs, respectively) was observed in all treatments, and irrespective to the leaf species (Fig. 4). This 20–45% increase in MDA content was led by its decrease that continued up to 3–6 h. Then, MDA augmentation was observed up to 48 h in the treatments with Cu and the mixtures of Cu and the extracts of both leaf species, however in different strength. Specifically, in the case of *A. glutinosa,* the addition of 100 mg/L DOC suppressed Cu-induced augmentation of MDA more strongly than the addition of 10 mg/L DOC (Fig. 4). At 48 h, no significant difference in MDA was observed between Cu and the combined treatment of Cu + 10-DOC$_{AG}$. Contrary, in the case of *A. negundo*, the addition of neither 10 nor 100 mg/L DOC were able to change significantly the course of Cu-induced augmentation of MDA (Fig. 4), the level of which, after 48 h, reached as almost twice higher as the control level.

After the initial approximately 20%-peak in MDA content at 1.5 h, the effect of 100 mg/L DOC extracts obtained from the NOM of both leaf species was weak, especially in the case of *A. glutinosa*, and showed no more than 15% deviation from the control level of MDA in exposures longer than six hours (Fig. 4).

## CAT activity

After 45 min exposure of *L. minor* plants, significant increases of CAT activities were observed in the treatments of Cu + 100-DOC$_{AG}$, Cu + 100-DOC$_{AN}$ and 100-DOC$_{AN}$ (Fig. 5). After 1.5 h, the plants responded by significant increases of CAT activities to Cu, Cu + 10-DOC$_{AG}$ and Cu + 100-DOC$_{AG}$ treatments in case of *A. glutinosa*, while the reaction in case of *A. negundo* was weaker. Then, after the initial increases in CAT activity,

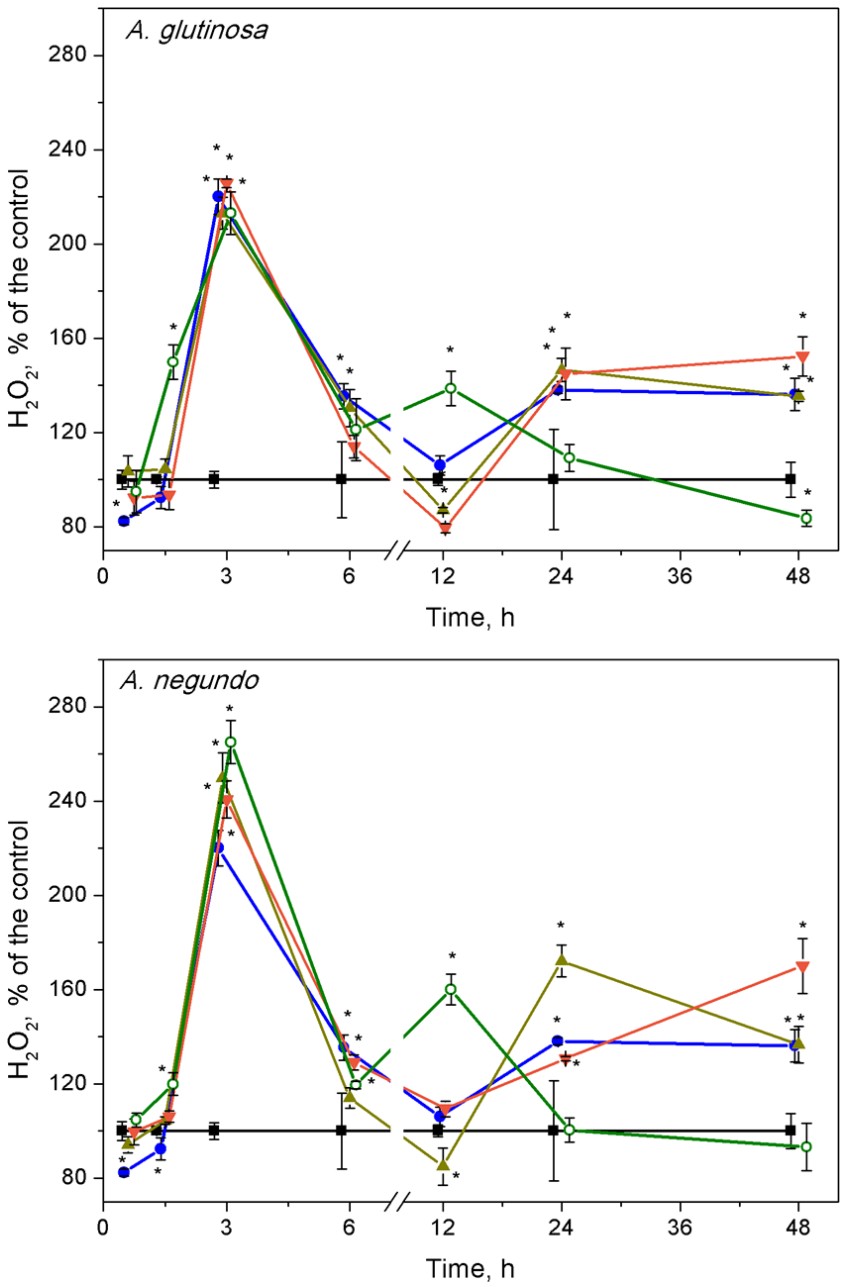

**Figure 3** **Concentration of hydrogen peroxide in *L. minor* treated by Cu, leaf litter extracts and Cu + leaf litter extracts.** Plants were incubated in control medium (■), 100 µM Cu (●), 100 mg/L DOC from *A. glutinosa* or *A.negundo* (o), and Cu+10 mg/L DOC (▼) or Cu + 100 mg/L DOC (▲). Each value represents mean ± SD ($n = 8$). Asterisks indicate significant difference from the control at $\alpha = 0.05$.

the reaction, in general, slowed down towards the control level ($682 \pm 188$ nkat/mg protein and $800 \pm 121$ nkat/mg protein for treatments of *A. glutinosa* and *A. negundo*, respectively; mean ± SD, $n = 56$) from 3$^{rd}$ hour and up to the end of exposure at 48 h, with irregular deviations. The exceptions comprised the decreasing tendency in CAT activities in the

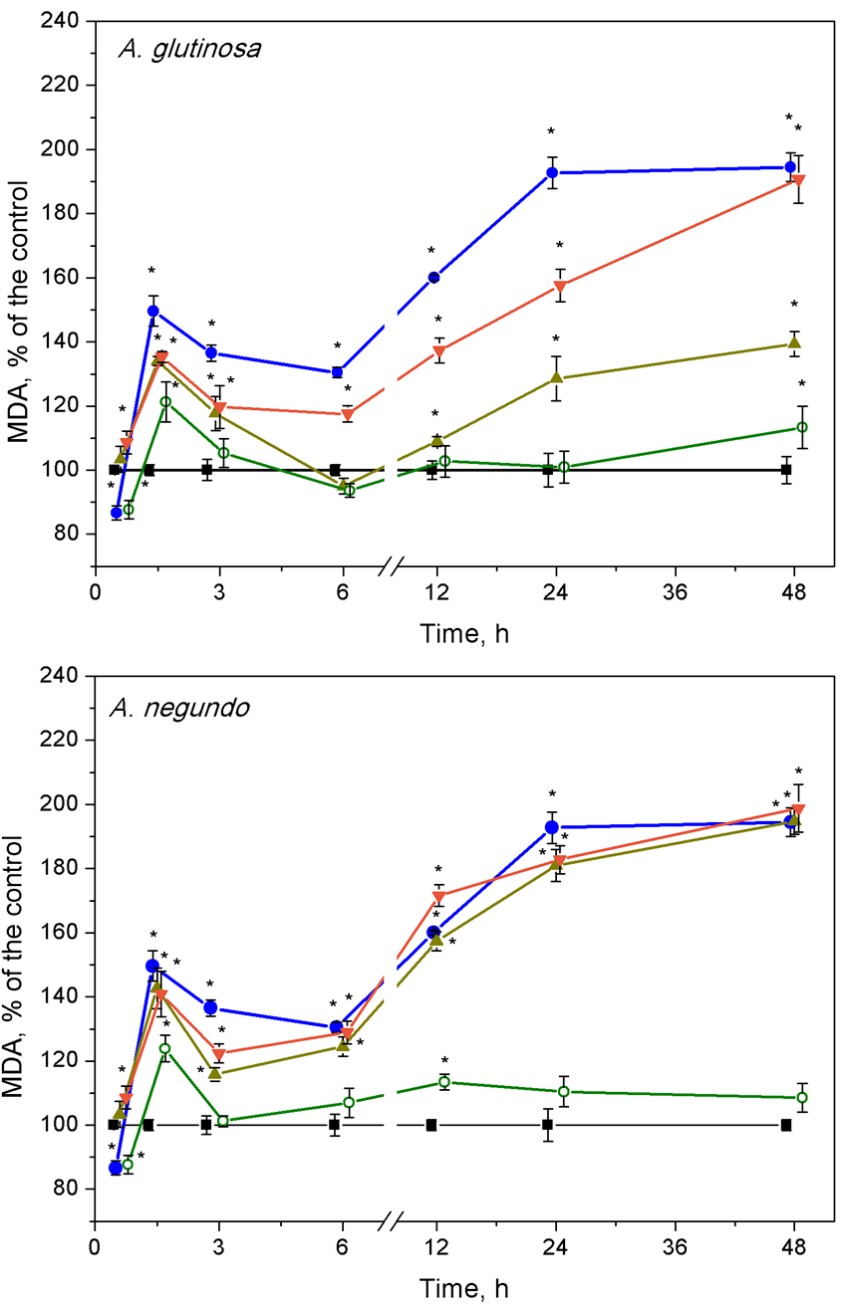

**Figure 4** **Lipid peroxidation expressed as MDA concentration in *L. minor* treated by Cu, leaf litter extract of *A. glutinosa* or *A. negundo*, and combinations of Cu and corresponding extracts.** Plants were incubated in control medium (■), 100 μM Cu (●), 100 mg/L DOC (o), and Cu + 10 mg/L DOC (▼) or Cu + 100 mg/L DOC (▲). Each value represents mean ± SD ($n = 8$). Asterisks indicate significant difference from the control at $\alpha = 0.05$.

treatments of 100 mg/L DOC of both leaf species and with Cu + 100-DOC$_{AN}$ reaching

significantly lower levels from those of controls by 20–30% at the end of exposure as well as the increasing tendency in CAT activities in the treatment of Cu + 10-DOC$_{AG}$ reaching significantly higher level from the controls by 20% at the end of exposure (Fig. 5).

### POD activity

Guaiacol peroxidase activities were significantly enhanced after treatments of Cu and its mixture with 10 and 100 mg/L DOC from *A. glutinosa*, but not with the 100-DOC$_{AG}$, during the 45 min–48 h period reaching a 2–2.5-fold increase at 24 and 48 h (Fig. 5). In case of *A. negundo*, the increases of POD activity over control (316 ± 122 nkat/mg protein and 406 ± 112 nkat/mg protein for treatments of *A. glutinosa* and *A. negundo*, respectively; mean ± SD, $n = 56$) were significant later, throughout the 12–48 h period, in the same treatments as above, and again with the exception of 100-DOC$_{AN}$ (Fig. 5).

### APX activity

No ascorbate peroxidase activity alterations were observed at 45th min, meanwhile, at 1.5 h, the 15–20% decreases were evident in the treatments of Cu + 10-DOC$_{AG}$ and Cu + 100-DOC$_{AG}$, and Cu + 10-DOC$_{AN}$ (Fig. 5). Then, at the 6–24 h period, the significant suppression of APX activities was observed in the treatments of Cu, Cu + 10-DOC$_{AG}$ and Cu + 100-DOC$_{AG}$, and at 6th hour, in the treatment of Cu + 10-DOC$_{AN}$ (Fig. 5). At the end of exposure at 48-h, the activities of all treatments did not differ from those of controls (32.6 ± 6.99 nkat/mg protein and 15.2 ± 2.47 nkat/mg protein for treatments of *A. glutinosa* and *A. negundo*, respectively; mean ± SD, $n = 56$).

### GR activity

The glutathione reductase activity changes in *L. minor* were irregular during the 12 h of exposure in all treatments with the DOM of *A. glutinosa* and Cu alone, then GR activity decreased by approximately 20% below control level (409 ± 46.6 nkat/mg protein and 430 ± 72.6 nkat/mg protein for treatments of *A. glutinosa* and *A. negundo*, respectively; mean ± SD, $n = 56$) in the treatments of Cu and Cu + 10-DOC$_{AG}$ (Fig. 5). At 24 and 48 h, GR activity increased up to approximately 20% in the treatment of Cu + 100-DOC$_{AG}$. In the case of *A. negundo*, the activity of GR activity was slightly, yet significantly, suppressed in the treatments of Cu + 10-DOC$_{AN}$ and Cu + 100-DOC$_{AN}$, during the 1.5–3-h period. Then GR activity tended to recover to the control level at 24 h and to exceed it at the end of 48 h exposure period, in the treatment of Cu + 10-DOC$_{AN}$, while a significant increase in 20–30% above control level was seen in the treatment of Cu + 100-DOC$_{AN}$ during the period of 12–48 h (Fig. 5). No significant alterations of GR activities were found in the LLE of 100 mg/L DOC of either species.

## DISCUSSION

Our results showed a high accumulation of copper in *L. minor*, which increased with time over 48 h. Similar accumulation properties of duckweed for this metal have already been documented for various exposure durations (*Drost, Matzke & Backhaus, 2007*; *Kanoun-Boulé et al., 2009*; *Razinger et al., 2007*). The accumulation of metals in aquatic plants is

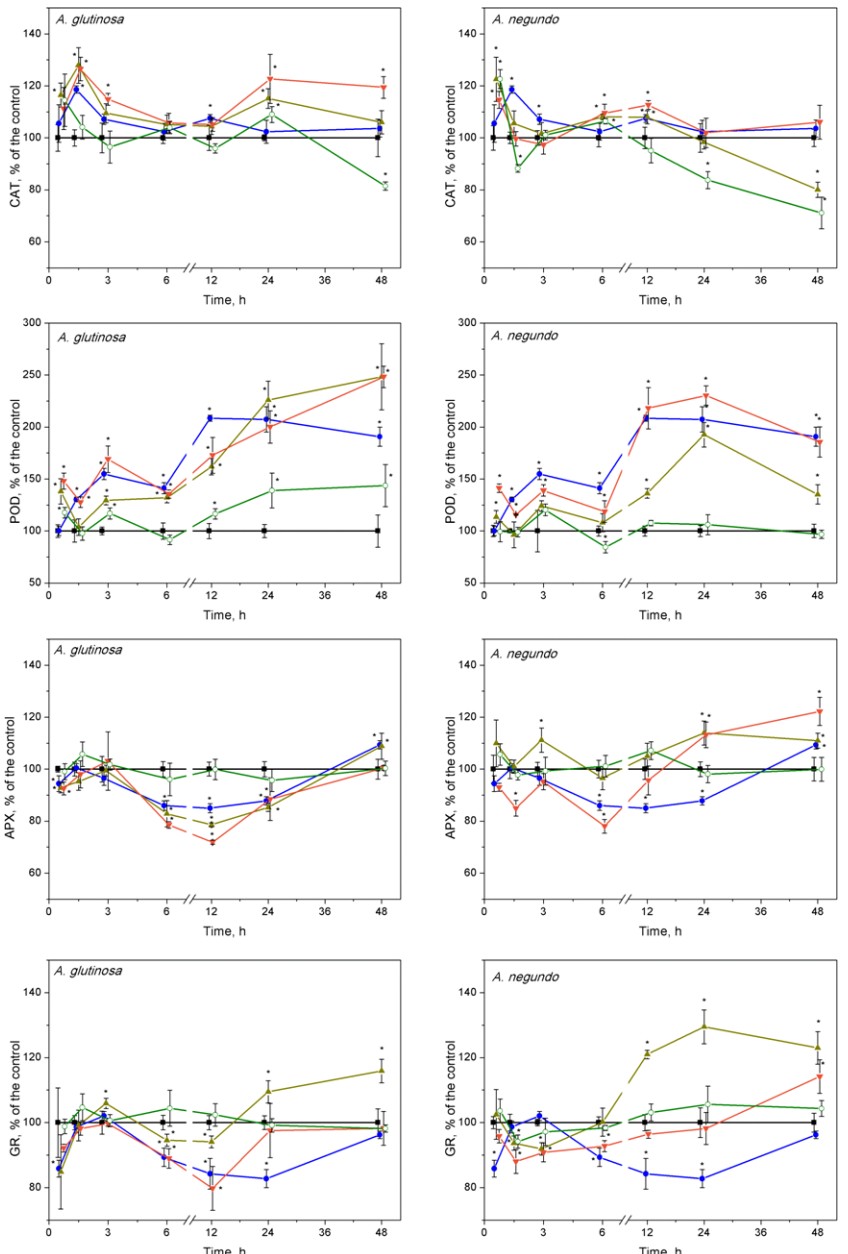

**Figure 5** **Oxidative stress enzyme activities in *L. minor* treated by Cu, leaf litter extract from *A. glutinosa* or *A. negundo*, and combinations of Cu and respective extracts.** Activities of catalase (CAT), guaiacol peroxidase (POD), ascorbate peroxidase (APX) and glutathione reductase (GR) in *L. minor* treated by Cu, DOC of leaf litter extracts from *A. glutinosa* or *A. negundo,* and combinations of Cu and the DOC of respective extract. Plants were incubated in control medium (■), 100 μM Cu (●), 100 mg/L DOC (o), Cu+10 mg/L DOC (▼) or Cu + 100 mg/L DOC (▲). Each value represents mean ±SD ($n = 8$). Asterisks indicate significant difference from the control at $\alpha = 0.05$.

<br>

often accompanied by a variety of morphological and physiological changes, some of which directly contribute to the tolerance capacity of plants (*Prasad et al., 2001*; *Xing, Huang & Liu, 2010*). *L. minor* exposed to Cu, the extracts from *A. glutinosa* or *A. negundo* leaf litter, and to the combination of Cu with the extracts indicated inhibition of the growth parameters, enhanced levels of the lipid peroxidation and altered enzyme activities. To explore oxidative damage, we applied certainly toxic concentration of Cu (100 µM) within up to 48-hour exposure durations as compared to growth inhibition values of 7-d EC50, i.e., 2.7 and 9.7 µM reported by Nauman et al. (2007) and Drost et al. (2007), respectively. The time span longer than about one day can already be considered as long-term since the amount of *L. minor* biomass doubles every two days (Environment Canada, 2007) and the doubling time for fronds ranges from 1.3 to 2.8 days (*Wang, 1990*) under optimal nutrient, light and temperature conditions. According to our findings, the reduction in root growth rate was higher than in the frond growth rate, irrespective of the kind of treatment. It is known that root length inhibition is a more sensitive endpoint than that of the frond area (*Gopalapillai, Vigneault & Hale, 2014*). However, the concentration-dependent antagonistic action of Cu and *A. glutinosa* extract mixture was evident on fronds and roots, while in the case of Cu and *A. negundo* extract mixture this was evident on roots only, suggesting that the origins of the diminishing influence of *A. glutinosa* and *A. negundo* extracts on Cu-induced toxicity effects should not be the same.

The complex interactions between physical-chemical and biological factors in the aquatic medium may change metal bioavailability (*Cuss & Guéguen, 2012*; *Koukal et al., 2003*). The data obtained in Cu measurements in *L. minor* within 45 min–48 h period revealed that the accumulation rate in the combined treatments of Cu and *A. glutinosa* or *A. negundo* leaf extracts (100 mg/L DOC) was respectively two or three times lower than that in the $CuSO_4$ treatment ($\sim$6 µg g$^{-1}$ FW h$^{-1}$). The Cu accumulation dynamics at a constant rate did not correspond with the free Cu ions concentration dynamics in media, wherein it reached relatively constant level after 1.5 h or 3 h, respectively for mixtures of Cu and *A. glutinosa* or *A. negundo* leaf extracts, and the $Cu^{2+}$ levels were respectively 39% and 51% of that of nominal Cu concentration. Fact that $Cu^{2+}$ concentration in the medium with NOM from *A. negundo* leaves remained higher than that from *A. glutinosa*, but a higher amount of Cu accumulated in *L. minor* in the case of NOM from *A. glutinosa* leaves suggests that part of Cu could enter the cell being bound on NOM or could be bound in the cell wall, and this part was higher in the case of *A. glutinosa* leaf extract. This assumption is supported by the comparative analysis of effects on *L. minor* root growth rate (RGR). As large as twice the increase of inhibition of RGR (= ($RGR_{Cu+100-DOC}$ - $RGR_{100-DOC}$)/$RGR_{100-DOC}$) was found for the mixture of Cu and *A. glutinosa* leaf extract as for Cu and *A. negundo* (Fig. 1). Otherwise, a stronger effect on *L. minor* exposed to the mixture of Cu and *A. negundo* leaf extract should be observed; at least, due to a higher concentration of Cu ions measured in the medium. The mechanisms of copper ions translocation from the medium into plant cell comprise, at least, transition metal transporters (*González-Guerrero et al., 2016*; *Palmgren & Nissen, 2011*) including low-molecular-weight organic molecules (*Sinclair & Krämer, 2012*) and binding in the cell wall. It was estimated that 74% of Cu accumulated

in the cell wall of macrophytic alga *Nitellopsis obtusa*, when its internodal cells were treated with 50 µM of $CuSO_4$ (*Manusadžianas et al., 2017*).

Plants exposed to long-term stress pass through different physiological states, from resistant to exhaustive (*Lichtenthaler, 1996*); thus, different kinetics of primary and successive plant responses might be expected. We found two phases of responses: namely, the first (up to 6 h) and the second one (within 6–48 h) could be identified for lipid peroxidation and hydrogen peroxide, and also, in part, enzyme activity kinetics. In addition, frond abscission/disintegration delimitated these response phases at the physiological level. Disintegration with successive chlorosis/necrosis is a recognized symptom of toxicity in fronds of *Lemna* exposed to pollutants (*Khellaf & Zerdaoui, 2009*). Release of daughter fronds from the metal stressed mother frond would increase the chance of the daughter fronds survival (*Li & Xiong, 2004*).

The high content of Cu (100 µgCu/g FW) accumulated in *L. minor* over initial 45 minutes caused a half-fold increase in the amount of MDA at 1.5 h and temporal increase of $H_2O_2$ with the peak at 3 h. It is widely accepted that malondialdehyde, the end product of lipid peroxidation, can be used as an indicator of membrane damage (*Heath & Packer, 1968*) and oxidative stress (*Zezulka et al., 2013*). At the end of the first response phase (plant resistance state), we observed the frond abscission. The oxidative stress progressed during the second response phase (plant exhaustive state) in terms of up to a 2-fold increase in MDA content and elevated $H_2O_2$ over control level at 48 h. A similar picture, in general, could be seen when plants were exposed to Cu and its combinations with leaf litter extracts; however, in the case of Cu and *A. glutinosa* leaf extract mixture, MDA content in *L. minor* was inversely dependent on DOC concentration in the extract (Fig. 4).

The response patterns of the plants exposed to each of the leaf litter extracts singly were mainly the same as exposed to Cu or Cu combined with the extracts, during the first response phase (up to 6 h). However, the levels of $H_2O_2$ and MDA relaxed to that of control during the second response phase. Therefore, it can be suggested that plants treated with extracts alone underwent oxidative stress in terms of $H_2O_2$ production and MDA increase, however, due to enzymatic activity they were able to resist this initial stress and consequently avoid detrimental effects in terms of growth and development at the final 7-day exposure, yet partially slowed down (Figs. 1 and 3 and 4).

In the first response phase, alterations of oxidative stress enzymes in *L. minor* were observed as early as after 45-min, the shortest exposure period. Specifically, Cu or its combinations with *A. glutinosa* extracts induced CAT activity increases that were not led by significant overrun of the control levels of $H_2O_2$ and MDA. This should show that the initial response of the enzymes prevented membrane damage due to ROS generation. Later on, however, concentrations of $H_2O_2$ and MDA significantly increased over the control reaching their peak values within 1.5–3 h, and this coincided with the maximum CAT activity. *Bayliak, Semchyshyn & Lushchak (2006)* and *Martins & English (2014)* have demonstrated that increasing $H_2O_2$ levels (up to 1mM) in yeast cells stimulate CAT. We measured $H_2O_2$ concentrations of 0.6–0.7 mM to be the highest ones in treated *L. minor*. Similar fast mobilization of CAT was considered as a cellular adaptation in primary leaves of *Phaseolus vulgaris* to cope with $H_2O_2$ overproduction generated by $Cu^{2+}$ (*Weckx &*

*Clijsters, 1996*). Interestingly, the increase in CAT activity during the initial 1.5 h was followed by attenuation of APX activity (Fig. 5). This might be linked with the decreased GR activity since GR converts GSSH into GSH by consuming NADPH and so maintains AsA level. One of the characteristic properties of APX that distinguishes it from POD, cytochrome c peroxidase and GR, is the rapid inactivation of the enzyme under conditions where an electron donor is absent (*Miyake & Asada, 1996*). The lower reduction state of AsA could be due to both the insufficient supply of electrons from GSH via direct interaction between GSH and AsA, and inadequate activities of MDHAR and DHAR (*Asada & Takahashi, 1987*; *Foyer & Halliwell, 1976*). In our experiments, GR activity began to increase after the initial decrease at 45 min, restoring the control level after three hours of exposure to Cu and Cu+DOC$_{AG}$ (Fig. 5). This increase of GR activity created, indirectly, favourable conditions for augmentation of ascorbic acid content, which allowed the APX activity to increase toward the control level during 1.5–3 h. Results of *Xiang & Oliver (1998)* have shown that the liquid culture of *Arabidopsis* tissues exposed to 100 μM Cu$^{2+}$ respond fairly rapidly by increasing the transcript levels of the genes encoding the GSH-synthesizing enzymes (GSH1, GSH2) and the GSSG-reducing enzyme (GR1). Elevated transcript levels were evident one hour after the exposure to Cu$^{2+}$, plateaued at six hours, and remained high for 18 h, and the high levels of transcripts continued for a few days under this condition. Another factor that prevented H$_2$O$_2$ accumulation in *L. minor* treated by Cu and Cu+DOC$_{AG}$ could be elevated POD activity within a 3–6-h period, which compensated gradual decrease of CAT activity toward the control level (Fig. 5). Similar concomitant activation of POD and inactivation of CAT has been found in various plants as an oxidative stress response to pathogen within 30 h (*Madhusudhan et al., 2009*).

The observed alterations in enzyme activities are likely related to oxidative reactions due to increased H$_2$O$_2$ level, which may eventually yield increased lipid peroxidation. Indeed, at the end of the first response phase, the highest MDA content in *L. minor* was found in the treatment with Cu (Fig. 2), which, however, was diminished by the DOC$_{AG}$, but not by the DOC$_{AN}$, in a concentration-dependent manner. The latter could be due to higher contents of polyphenols in the DOC of *A. glutinosa* than that of *A. negundo*, respectively 49 and 23 mg PAE/g DM and, in addition, tannins (7.5 mg PAE/g DM) that were lacking in the DOC of *A. negundo*. It has been found that tannins exhibit strong antioxidant properties in comparison to low molecular weight phenolic compounds. The presence of catechol or galloyl groups in tannin structure are essential to complex formation with transition metals (*Andjelkovic et al., 2006*) including copper (*Brown & Kelly, 2007*; *Miller et al., 1996*). Phenolic compounds with additional hydroxy groups on aromatic ring bind Cu$^{2+}$ more efficiently (*Brown & Kelly, 2007*). The lack of tannins and low polyphenol content, and thus weak Cu binding ability in *A. negundo* LLE may explain, at least in part, negligible impact of *A. negundo*-derived DOM on the moderation of Cu-induced MDA levels. However, it is also known that phenolics can display prooxidant activities in the presence of metal ions in plants (*Decker, 1997*; *Azam et al., 2004*).

In the second response phase, Cu and its combination with *A. glutinosa* extracts induced a continuous increase of MDA concentration up to the end of 48-h exposure. This could be related to inactivation of APX, despite that the CAT and POD activity remained above the

control level. It is known that inhibition of APX results in an increased level of $H_2O_2$ that contributes to defence gene activation and acts as a substrate for POD involved in defence responses such as lignification and crosslinking of cell wall proteins (*Bradley, Kjellbom & Lamb, 1992*). DOC of *A. glutinosa* is characterized by higher contents of polyphenols and tannins that are known to support the primary detoxification system (*Yamasaki, Sakihama & Ikehara, 1997*). Our results also showed that *A. glutinosa* leaf extract stimulated POD when it acted individually. In the Cu+DOC$_{AG}$ treatments, exogenous polyphenols and tannins and continuous increase of POD activity, exceeding that of Cu-induced at the end of the 48-h period, suggests that *L. minor* avoids $H_2O_2$ overproduction through a phenolic-dependent resistance mechanism. This possibility is further supported by the observation that MDA content was diminished by the DOC$_{AG}$ in a concentration-dependent manner, which was also true for the first response phase.

Combined treatments of Cu and *A. negundo* extracts, within initial 3 h, induced different responses of certain oxidative stress enzymes in relation to those of Cu and *A. glutinosa* extracts. Contrary to the case of *A. glutinosa*, when CAT enzyme activity decreased monotonically toward controls (the tendency observed up to 6 h), a rapid decrease of CAT activity to control level at 1.5 h indicated that the ability to scavenge $H_2O_2$ in plants was weakened. Indeed, *A. negundo* extract (100 mg DOC/L) even acting individually induces fast inactivation of CAT. During 3–6 h period, the relatively higher activity of POD in the relation of CAT, APX and GR activities in the treatments of Cu and *A. negundo* extracts indicated that actual level of lipid peroxidation could be mainly associated with the POD scavenging of $H_2O_2$. Inactivation of both CAT and the AsA-GSH pathway may prevent the cell from depleting NAD(P)H reserves (*Heineke et al., 1991*). Overall, within three hours of exposure, the treatments of Cu in combination with DOC from either *A. negundo* or *A. glutinosa* extract yielded lower MDA levels than in the treatment of Cu, and this was supported by various responses of ROS enzymes. However, after six hours of exposure, plants were unable to cope with $H_2O_2$ excess in the treatments of Cu+DOC$_{AN}$ when lipid peroxidation augmented up to the level observed in Cu-treatment, in contrast to Cu+DOC$_{AG}$ when lipid peroxidation was lower (Fig. 4). This distinction between the influences of various leaf species extracts at the background of Cu action could be caused by the differences in contents of polyphenolic compounds (see above).

Extended exposure of *L. minor* for 6–48 h in the mixture of Cu+100-DOC$_{AN}$ induced higher activities of APX and GR. This finding indicates that consumption of $H_2O_2$ can be associated not only with a POD (see below), but with the enzymatic activities in the AsA-GSH cycle, as well. Alteration of these enzymes was led by the inactivation of CAT. The first reason for the latter could be an excess level of AsA in the presence of Cu. *Davison, Kettle & Fatur (1986)* have shown that AsA alone is not very damaging and that AsA's inhibitory action can be released by $O_2$, $H_2O_2$ or $Cu^{2+}$, i.e., when AsA is oxidized to semidehydroascorbate (or ascorbyl radical). Ascorbate toxicity depends on the presence of copper (or iron) and oxygen, but oxygen is not required in the presence of $H_2O_2$ (*Samuni et al., 1983*). Another reason for progressive inactivation of CAT observed in our study could be the presence of phenolic compounds in leaf litter extracts. Although the quantity and composition of the extracts differed, the effect on CAT activity decrease was evident

in 100 mg/L DOC of either extract, especially at the end of 48-h exposure. The similar inhibitive action of certain phenolic compounds on CAT within 24-h exposure has been found for thermophilic fungi (*Yüzügüllü et al., 2011*).

POD activity did not differ between the treatments of Cu and either Cu+DOC$_{AN}$ or Cu+DOC$_{AG}$ at 24th hour of exposure. However, the extract type probably determined the opposing POD response at 48th hour of exposure. At this time point, the highest POD activity (exceeding that of the Cu-induced) was observed in the case of Cu+DOC$_{AG}$. The lack of the differences in MDA levels in the second phase of *L. minor* response to Cu or its combinations with DOC$_{AN}$ suggests that, under the influence of *A. negundo* extracts, CAT, POD and AsA-GSH cycle enzymes were unable to minimize the oxidative damage induced by Cu. This is the opposite to the action of *A. glutinosa* extracts.

Invasive plant species tend to migrate from their native habitats under favourable climatic conditions; therefore, trophic and other relationships in ecosystems are changing. It has been suggested that black alder *Alnus glutinosa*, native species in Lithuania, and boxelder maple *Acer negundo*, invasive species in Lithuania, impact the same aquatic organisms in different ways (*Krevš et al., 2013*; *Manusadžianas et al., 2014*). In this context, we revealed the potential of diverse species DOM to modify Cu toxicity effects on *Lemna minor*. Both types of leaf litter extracts protected *L. minor* from deleterious effects of lipid peroxidation products during the first response phase (up to 6 h) when plants activated stress-coping mechanisms. Throughout the second response phase (6–48 h), cellular defence mechanisms were impaired and the vitality of *L. minor* steadily decreased. Overall, the analyses of Cu accumulation in *L. minor* and binding on DOM, and the dynamics of MDA content that represents integrative biochemical response, suggest that the reason of beneficial action of *A. glutinosa* extracts compared to that of *A. negundo* is based on the higher contents of polyphenols and tannins.

## CONCLUSIONS

We revealed that leaf litter extracts of black alder *Alnus glutinosa*, native species in Lithuania, and boxelder maple *Acer negundo*, invasive species in Lithuania have various potential to modify Cu toxicity effects on *Lemna minor*. Analyses of duckweed responses, dynamics of Cu accumulation in the plant and its binding on the DOM in media allowed to conclude that both types of leaf litter extracts protected *L. minor* from deleterious effects of lipid peroxidation products during the first response phase within 6 h, however, cellular defence mechanisms were impaired during the prolonged exposure within 6–48 h. The differences in antioxidant enzyme activity profiles ascertained in *L. minor* treated by mixtures of Cu and various leaf extracts over two days were considered to condition dissimilar effects on the development of plant fronds and roots observed after seven days. The complex data obtained in the current study could be useful for modelling of aquatic ecosystem responses to the changing environment.

## ACKNOWLEDGEMENTS

We thank Dr Rimantas Vitkus for discussing data statistical analysis.

### Funding

The authors received no funding for this work.

### Competing Interests

The authors declare there are no competing interests.

### Author Contributions

- Rolandas Karitonas conceived and designed the experiments, performed the experiments, analyzed the data, prepared figures and/or tables, authored or reviewed drafts of the paper, and approved the final draft.
- Sigita Jurkonienė analyzed the data, authored or reviewed drafts of the paper, and approved the final draft.
- Kazys Sadauskas conceived and designed the experiments, authored or reviewed drafts of the paper, and approved the final draft.
- Jūratė Vaičiūnienė performed the experiments, prepared figures and/or tables, and approved the final draft.
- Levonas Manusadžianas conceived and designed the experiments, analyzed the data, authored or reviewed drafts of the paper, and approved the final draft.

### Data Availability

Raw data are available in the Supplementary Files.

### Supplemental Information

Supplemental information for this article can be found online at http://dx.doi.org/10.7717/peerj.9444#supplemental-information.

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
