# Peer review of "Modifying effects of leaf litter extracts from invasive versus native tree species on copper-induced responses in Lemna minor"

_PeerJ, doi:10.7717/peerj.9444_

## Round 0.1 · original submission · Minor Revisions

The reviewers have highlighted the strengths of your submitted manuscript and have proposed a number of changes to improve the quality. Please consider the proposed changes as mandatory for a final acceptance of your manuscript.

While some comments address more technical aspects such the reference list (with a number of missing references as well as references not cited in the text), optimisation of figures (specifically Fig. 5) and the improvement of the language, there are also content-related aspects to be resolved such as a more comprehensible title and a more targeted abstract. Furthermore, as indicated by reviewer 3, you should discuss in more detail the possible role of reduced metal bioavailability of DOM-/DOC-metal-complexes as a cause for the reduced metal toxicity in your assay.

I look forward to your revised manuscript.

Reviewer 1 ·

Basic reporting

Literature references, sufficient field background / context provided, but
check the literature list.
Not in reference list: González-Guerrero et al., 2016; Palmgren and Nissen, 2011; Sinclair and Krämer, 2012; Khellaf and Zerdaoui, 2009, Madhusudhan et al., 2009.

Check the literature cited carefully. There are 19 items in the references, which are missing from the text.

Drawings are legible except Fig. 5 in which the line markings are hardly visible.

Experimental design

Original primary research within Aims and Scope of the journal.
The manuscript refers to global trends in environmental protection and exploring the possibilities of limiting the toxicity of heavy metals (in this case copper) by extracts from leaf mulch from trees of various species. The authors aptly used Lemna minor, which is a recognized natural indicator of toxicity in the world. In addition, a large number of physiological studies deserve recognition, which show the reactions of plants (Lemna minor) to copper and the possibility of reducing its toxic effect by extracts from the leaves of the studied tree species.
Research question well defined, relevant & meaningful. It is stated how research fills an identified knowledge gap.
Rigorous investigation performed to a high technical & ethical standard.
Methods described with sufficient detail & information to replicate.

Validity of the findings

All underlying data have been provided; they are robust, statistically sound, & controlled.
Conclusions are well stated, linked to original research question & limited to supporting results.

Additional comments

Line 1-2 Change the title to a more understandable one
Line 15-44 In Abstract, the research background, the purpose of the work, methods, results without providing unnecessary details and conclusions should be more demonstrated. The current form has a lot of details that take up a lot of space and make understanding difficult
Line 99 - should be supplemented with cyanobacterial toxicity (Literature)
Line 18-29 The data presented should be described in a more understandable way
In the Results chapter, when describing the response of individual parameters of physiological activity, it would be beneficial to insert at the beginning a general sentence about the effect of treatment on this parameter. Then you can document it with detailed results just as the authors did.
The discussion is written properly.
Drawings are legible except Fig. 5 in which the line markings are hardly visible.
Not in reference list: González-Guerrero et al., 2016; Palmgren and Nissen, 2011; Sinclair and Krämer, 2012; Khellaf and Zerdaoui, 2009, Madhusudhan et al., 2009.

Check the literature cited carefully. There are 19 items in the references, which are missing from the text.

Line 477-482 - The text is repeated in 483-488.

The manuscript should be linguistically corrected, since many sentences are too long, making it difficult to understand the text.
The results obtained are interesting, bring a lot of hitherto unknown information and therefore, after applying the proposed corrections, they should be published.

Reviewer 2 ·

Basic reporting

The manuscript concerns the exploring of the possibilities of reducing copper toxicity by leaf mulch extracts from tree two species using Lemna minor as an indicator of toxicity. The effect of litter extracts and copper as assessed on the basis of many parameters of physiological activity in Lemna minor. The work is interesting and brings many new elements.
The manuscript requires proofreading before publication.
The title should be shortened to make it easier to understand
In Abstract it contains a lot of details that take up a lot of space and make understanding difficult. Its content should articulate research background, the purpose of the work, methods, results and conclusions.
In the Results chapter, it would be beneficial to present an additional conclusion on the impact of treatments on individual assessment parameters. Currently, only detailed data is often provided on the basis of which the reader must guess the existing relationships
The drawings are legible except for Fig. 5 in which the line markings are hardly visible.
The captions under the drawings are too long.
Not all literature references cited in the text are in the reference list and vice versa
The manuscript should be linguistically corrected,
The results obtained bring a lot of new, previously unknown information from a scientific point of view and after the proposed correction of the text they should be published.

Experimental design

The research methodology is carried out correctly in accordance with international standards

Validity of the findings

The results obtained bring new information previously unknown in the literature. Noteworthy is the large number of physiological tests performed, which indicate Lemna minor's reactions to the treatments used

Additional comments

The manuscript may be published after the adjustment. Comments are given above.

Reviewer 3 ·

Basic reporting

The English needs work; in many places, there are too many citations to support a single statement - it's not clear whether they each portray a different aspect of the statement or why so many are needed; structure is fine; figures are fine.

Experimental design

I think the basic question is whether DOC from invasive vs native species alter Cu toxicity to Lemna, and this would be additional reason to limit invasive species. In that case, the experimental design is appropriate.

Validity of the findings

The effect of DOC or DOM on dissolved cationic metal toxicity to aquatic organisms, such as Lemna, is reasonably well-studied. DOM or DOC has been included in a number of regulatory Biotic Ligand Models for metals. Research has also been conducted on how various different sources of DOC have different modifying effects on metal toxicity - relative to their affinity for the metal thus creating a metal-complex that is less or not bioavailable. This might be in play in this study, not sure - but the authors should talk about this. Generally, the very great detailed biochemistry assays are interesting, but the link to toxicity endpoints is less obvious.

---

## Round 0.2 · accepted · Accept

Thank you very much for your thorough revision of the manuscript and the consideration of all reviewer's comments.

Reviewer 1 ·

Basic reporting

Enter 'no comment' if you have nothing to add.

Experimental design

Enter 'no comment' if you have nothing to add.

Validity of the findings

Enter 'no comment' if you have nothing to add.

Additional comments

Enter 'no comment' if you have nothing to add.

Reviewer 2 ·

Basic reporting

No additional comment to the previous review

Experimental design

No additional comment to the previous review

Validity of the findings

The results obtained bring a lot of new, previously unknown information from a scientific point of view and in its current form it should be published.

Additional comments

In the revised manuscript submitted for re-review, all my comments were taken into account. That is why I recommend it for publication.